# LLM Internal States Reveal Hallucination Risk Faced With a Query

**Ziwei Ji, Delong Chen, Etsuko Ishii, Samuel Cahyawijaya,**
**Yejin Bang, Bryan Wilie, Pascale Fung**
Center for Artificial Intelligence Research (CAiRE)
Hong Kong University of Science and Technology
zjiad@connect.ust.hk, pascale@ece.ust.hk

## Abstract

The hallucination problem of Large Language Models (LLMs) significantly limits their reliability and trustworthiness. Humans have a self-awareness process that allows us to recognize what we don't know when faced with queries. Inspired by this, our paper investigates whether LLMs can estimate their own hallucination risk before response generation. We analyze the internal mechanisms of LLMs broadly both in terms of training data sources and across 15 diverse Natural Language Generation (NLG) tasks, spanning over 700 datasets. Our empirical analysis reveals two key insights: (1) LLM internal states indicate whether they have seen the query in training data or not; and (2) LLM internal states show they are likely to hallucinate or not regarding the query. Our study explores particular neurons, activation layers, and tokens that play a crucial role in the LLM perception of uncertainty and hallucination risk. By a probing estimator, we leverage LLM self-assessment, achieving an average hallucination estimation accuracy of 84.32% at run time.[1]

## 1 Introduction

Humans have an awareness of the scope and limit of their own knowledge (Fleming and Dolan, 2012; Koriat, 1997; Hart, 1965), as illustrated in Fig. 1. This cognitive self-awareness ability in humans introduces hesitation in us before we respond to queries or make decisions in scenarios where we know we don't know (Yeung and Summerfield, 2012; Nelson, 1990; Bland and Schaefer, 2012). However, LLM-based AI assistants lack this cognitive uncertainty estimation. Consequently, they tend to be overconfident and may produce plausible-sounding but unfaithful or nonsensical contents called *hallucination* or *confabulation* (Ji et al., 2022; Xiao and Wang, 2021; Bang et al., 2023; Xiong et al., 2023). This problem limits their

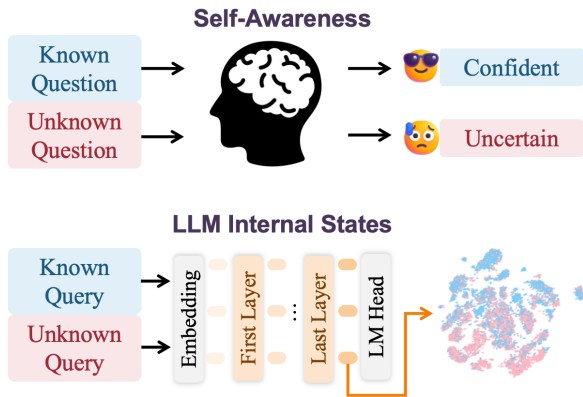

Figure 1: Humans have self-awareness and recognize uncertainties when confronted with unknown questions. LLM internal states reveal uncertainty even before responding. Pink dots are the internal LLM states associated with hallucinated responses, whereas Blue dots are those of faithful responses. The queries leading to those LLM responses are colored accordingly.

applications in numerous real-world scenarios and undermines user trustworthiness.

Previous research (Bricken et al., 2023; Templeton et al., 2024; Bills et al., 2023; Wu et al., 2024) have explored the internal states of language models that capture contextual and semantic information learned from training data (Liu et al., 2023; Chen et al., 2024; Gurnee and Tegmark, 2023). Nevertheless, internal states of language models sometimes exhibit limited generalization on unseen data and their representation effectiveness can be undermined by flawed training data or modeling issues (Wang et al., 2022a; Belinkov and Glass, 2019; Meng et al., 2021; Xie et al., 2022; Carlini et al., 2021; Yin et al., 2023a). Notably, recent works have shown that the LLM's internal states can potentially detect hallucinations in texts (Azaria and Mitchell, 2023; Chen et al., 2024; Su et al., 2024). However, these works examine texts not exclusively produced by the same LLMs whose internal states are analyzed, highlighting the necessity for further investigation into the LLM *self-awareness*

---

[1] The source code can be obtained from https://github.com/ziweiji/Internal_States_Reveal_Hallucination

and how their internal states correlate with their uncertainty and *own* hallucination occurrence.

Our work takes a step further by investigating **whether LLM internal states have some indication of hallucination risk given queries and whether it can be reliably estimated even before the actual response generation** (Fig. 1). We conduct a comprehensive analysis of LLMs internal mechanisms in terms of training data sources and across 15 diverse NLG tasks that extend beyond the QA task (Snyder et al., 2023; Slobodkin et al., 2023) and span over 700 datasets. We explore particular neurons, different activation layers, and tokens that play a crucial role in the LLM perception of uncertainty and hallucination risk. Employing a probing estimator (Belinkov, 2022) on the internal states associated with the queries, we validate their self-awareness and ability to indicate uncertainty in two aspects: (1) Whether they have seen the query in training data, achieving an accuracy of 80.28%. (2) Whether they are likely to hallucinate regarding the query, achieving an average estimation accuracy of 84.32% across 15 NLG tasks. We propose that understanding these representations could offer a proactive approach to estimating uncertainty, potentially serving as an early indicator for the necessity of retrieval augmentation (Wang et al., 2023) or as an early warning system.

## 2 Hallucination and Training Data

The sources of hallucination in LLMs can be traced back to *data* and *modeling* (Ji et al., 2022). Factors tied to data encompass unseen knowledge, task-specific innate divergence, noisy training data, etc. From the modeling perspective, hallucinations can be traced to the model architecture, alignment tax, teacher-forced maximum likelihood estimation (MLE) training, etc.

A common situation where hallucinations from *data* occur is when LLMs attempt to provide information on **unseen** queries that are not included in their training set, rather than refusing to reply. Previous works (Kadavath et al., 2022; Rajpurkar et al., 2018; Onoe et al., 2022; Yin et al., 2023b) explore to identify the unseen data based on various indicators such as text similarity, perplexity. We investigate the capability of LLMs to recognize **whether they have seen the query in training data** via novel analysis of their internal states. To facilitate analysis, we craft two sets of queries by collecting news from periods *before* and *after* the

release of the LLM we analyze to represent unseen and seen data, respectively.

However, in the real-world scenario, it's impractical to definitively categorize data as entirely seen or unseen due to the inability to access the vast training data of LLM. Thus, we expand the preliminary insights and further investigate LLMs' self-awareness of recognizing **whether models are likely to hallucinate regarding the query.** It's important to note that the hallucinations are *source-agnostic*, meaning they can result from both unseen and seen data. The latter can still trigger hallucinations due to deficiencies in *modeling*. To facilitate analysis, we construct data by using LLM to directly generate responses to queries across diverse NLG tasks and then label the hallucination level in the responses.

## 3 Methodology

This section begins with an introduction to the problem formulation of uncertainty estimation faced with queries in § 3.1. We construct datasets in § 3.2 focusing on two dimensions: (1) the distinction between queries seen and unseen in the training data; (2) the likelihood of hallucination risk faced with the queries. To validate the efficacy of internal state representation in hallucination estimation, we visualize the neurons for perception extracted from a specified LLM layer (§ 3.3) and then leverage the *probing classifier* technique (Belinkov, 2022) on top of internal states associated with the last token of queries (§ 3.4).

### 3.1 Problem Formulation

Suppose we have an LLM $f$ parameterized by $\theta$. It is able to gain internal states $I$ and generate response $r$ given user query $q$ represented as $I_{\theta,q}, r_{\theta,q} = f_\theta(q)$. We aim to investigate the *self-awareness* of LLM, specifically how their internal states $I$ relate to their level of hallucination risk $h$ when faced with a query $q$.

We employ a dataset $\mathcal{D}_\theta = \{\langle I_{\theta,q,i}^{\text{train}}, h_i^{\text{train}} \rangle\}_{i=1}^N$ consisting of $N$ query-label pairs. These pairs serve to represent the behavior of $f_\theta$ Here, $h_i^{\text{train}}$ denotes the level of hallucination risk, which is labeled based on (1) the query's presence in the training data or (2) the degree of hallucination in the response $r$ to $q$. Thus, our objective is mathematically expressed as:

$$h = \mathbb{E}(I_{\theta,q}; \mathcal{D}_\theta) \qquad (1)$$

Here, $\mathbb{E}$ signifies an estimator function. The combi-

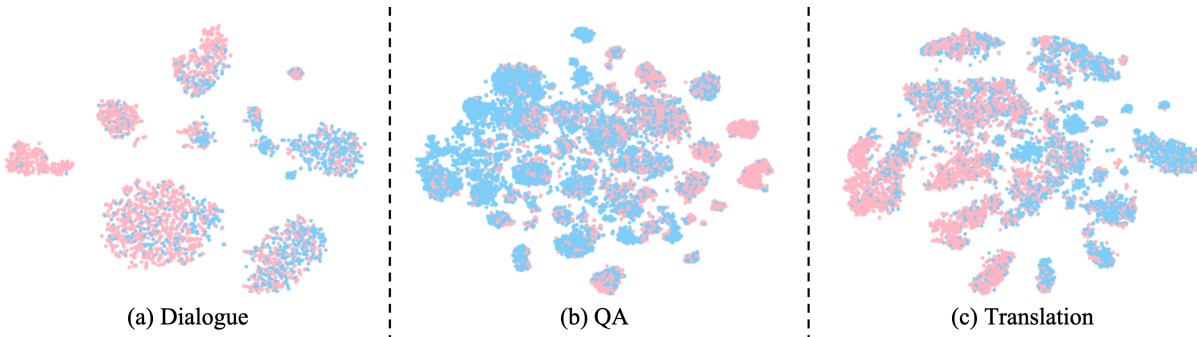

| (a) Dialogue | (b) QA | (c) Translation |

Figure 2: Visualization of the Neurons for Hallucination Perception in various NLG tasks. Pink dots represent Unknown Queries triggering hallucinations and Blue dots represent Known Queries.

nation of *model-specific* attribute via $f_\theta$, and *query-only* attribute via $q$, allows it to accurately capture the characteristics of individual LLMs and fosters a more efficient prediction mechanism that mirrors human cognitive processes.

## 3.2 Data Construction

As introduced in § 1 and 2, we investigate LLM internal states' self-awareness and ability to indicate uncertainty in two aspects: (1) whether they have seen the query in training data; and (2) whether they are likely to hallucinate when faced with the query.

**(1) Seen/Unseen Query in Training Data** The unseen queries will trigger hallucinations due to the lack of information within the model's training data when the model doesn't refuse to respond. In other words, hallucinations triggered by unseen queries are *data*-related. To investigate the distinguishability between seen and unseen queries, we construct a compact dataset consisting of two distinct sets of queries. For the seen group, we utilize historical BBC news in 2020 highly likely exposed during LLM's training. For the unseen group, we utilize recent BBC news in 2024 *after* the release of the LLM we analyze. To ensure comparability, we ensure these two sets share similar length distributions and semantic information via sentence embeddings [2]. The two groups are both from LatestEval (Li et al., 2024), a benchmark designed to tackle data contamination in evaluation through dynamic and time-sensitive construction. The query is `Tell more details about the news: {news_title}`.

**(2) Hallucination Risk faced with the Query**
We first construct data using LLM to directly generate responses to queries in diverse NLG tasks. Subsequently, for labeling the responses and corresponding queries, a comprehensive integration of NLG metrics assesses the levels of hallucination.

We select 15 NLG task categories including QA, Summarization, Translation, etc consisting of over 700 datasets from **Super-Natural Instructions** benchmark (Wang et al., 2022b) [3]. This generation only uses the parametric knowledge of LLM which is a proxy of performance in real-world applications. This generation process can also be in other settings, such as retrieval-augmented generation (RAG), to explore whether the internal states can estimate hallucination risk or other aspects' performance in these settings.

To evaluate the generated responses, we implement a multi-faceted evaluation approach. We employ classical **Rouge-L** (Lin, 2004), which compares the generated response with gold-standard reference. To measure hallucination level, we also utilize **Natural Language Inference (NLI)** [4] and **Questeval** (Scialom et al., 2021). **NLI** is a common metric for hallucination evaluation (Ji et al., 2023a,b) which assesses the logical consistency/entailment of generated text with the provided context or the reference. **QuestEval** is a QA-based metric for evaluating the faithfulness of the output in generation tasks. This work adopts its reference-dependent mode depending both on the input source and golden reference.

To make up for deficiencies of single automatic metrics, we integrate these three metrics compre-

---

[2] https://huggingface.co/sentence-transformers/all-mpnet-base-v2

[3] Please find the full list of NLG task categories in Fig. 3 or 4 and the full list of tasks in Tab. A2 in § A.

[4] https://huggingface.co/MoritzLaurer/mDeBERTa-v3-base-xnli-multilingual-nli-2mil7

hensively. If NLI predicts entailment and both Rouge-L and Questeval exceed their respective median values, we assign a label of **1**. Conversely, if NLI predicts contradiction or neutrality, and both Rouge-L and Questeval fall below their median values, we assign a label of **0**. This labeling strategy not only provides a binary quality assessment but also reflects a multi-dimensional evaluation of the text, capturing the hallucination level of the generated responses.

### 3.3 Preliminary Analysis: Neurons for Hallucination Perception from Internal States

Internal states play a crucial role in language models, encapsulating rich contextual and semantic information learned from predicting tokens. They are adept at recognizing complex patterns and relationships pertinent to various NLP tasks, which positions them as potentially powerful tools for estimating the risk of hallucinations (Azaria and Mitchell, 2023; Liu et al., 2023; Chen et al., 2024). Furthermore, previous works (Azaria and Mitchell, 2023; Ahdritz et al., 2024; Liu et al., 2024) have demonstrated that the activations of the last token from the last layer in LLMs contain one of the most useful features. Therefore, we take these representations for preliminary analysis on the self-assess sense of internal states and the role of specific neurons in the uncertainty and hallucination estimation. Specifically, we employ a feature selection method based on Mutual Information (Kraskov et al., 2004) to measure the relevance of different features/dimensions for distinguishing between the categories in a dataset.

In the context of NLG tasks including dialogue, QA, and translation, we select the eight most significant neurons/dimensions from the last activation layer and visualize them in Fig. 2. We observe that these neurons exhibit sensitivity to uncertainty, allowing them to distinguish between different hallucination levels given known and unknown queries. In other words, there exist individual neurons within LLM that can fairly perceive uncertainty and predict future hallucinations. This approach not only enhances our understanding of the neural correlates of hallucinations but also paves the way for developing targeted interventions that mitigate the effects of hallucinations.

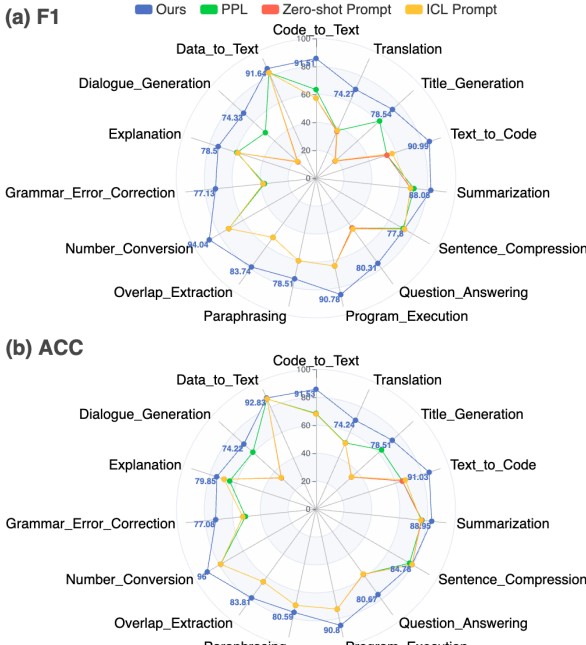

Figure 3: Automatic evaluation results for our method and baselines including Perplexity (PPL), Zero-shot Prompt, and In Context Learning (ICL) Prompt.

### 3.4 Internal State-based Estimator

Based on the above preliminary analysis and previous works, we use the activations corresponding to the last token of queries from a specified layer in LLMs, denoted as $x_q$, as the input for our estimation model. The accessibility and ease of obtaining these states further underscore their practicality for such applications.

For the architecture of our estimator, we employ a variant of the multilayer perceptron (MLP) adapted from the Llama (Touvron et al., 2023). The estimator is mathematically formulated as:

$$H = \text{down}(\text{up}(x_q) \times \text{SiLU}(\text{gate}(x_q))) \quad (2)$$

where $\text{SiLU}$ is the activation function. $\text{down}$, $\text{up}$, and $\text{gate}$ are linear layers for down-projection, up-projection, and gate mechanisms, respectively. The combination of internal states and the Llama MLP structure handles the complexity of hallucination risk estimation in NLG tasks.

## 4 Experiments

### 4.1 LLM

In this work, we primarily use Llama2-7B (Touvron et al., 2023) as our generative model and delve into its internal states to access hallucination risk estimation. In addition, we explore the impact of

different internal states in the Mistral-7B (Jiang et al., 2023) in § 5.

## 4.2 Baselines

To explore query-only uncertainty estimation, we involve straightforward prompt-based approaches as baselines.

**Zero-shot Prompt** We directly ask the LLM whether it can accurately respond to the query via the following prompt: "Query: {Query}$\backslash n\backslash n$Are you capable of providing an accurate response to the query given above? Respond only to this question with 'yes' or 'no' and do not address the content of the query itself."

**In-Context-Learning (ICL) Prompt** We ask the LLM whether it can accurately respond to the query and give some examples: "Are you capable of providing an accurate response to the following query? Respond only to this question with 'yes' or 'no' and do not address the content of the query itself.$\backslash n\backslash n$Query: {Example Query 0}$\backslash n$Answer: no$\backslash n\backslash n$Query: {Example Query 1}$\backslash n$Answer: yes...$\backslash n\backslash n$Query: {Query}$\backslash n$Answer:"

**Perplexity (PPL)** Considering the prompt-based methods only use the model's inner knowledge, we also incorporate the distribution of the training dataset and employ a Perplexity (PPL)-based baseline. Assume LLMs are trained on a hypothetical large dataset that perfectly contains every possible query-response pair, where the responses are guaranteed to be faithful. Then, the hallucination estimation can be simply done by checking whether the given query appears in the training corpus (Lee et al., 2021; Kandpal et al., 2023). To determine this threshold, we first calculate the PPL for each query. Subsequently, we identify the optimal PPL threshold that yields the maximum accuracy on our training dataset. This optimal threshold is then applied to the test dataset to gauge the accuracy of our hallucination risk estimation method.

## 4.3 Estimator Evaluation Protocols

For the classification task with the discrete type predicted, we utilize **F1** and **Accuracy** to measure the quality of predicted categorization.

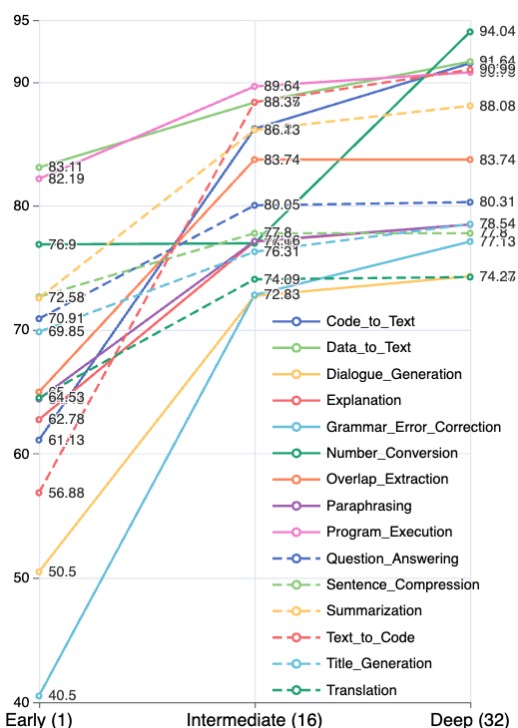

Figure 4: F1 scores of Internal-State from Different Layers for Hallucination Estimation.

| Training Task | Testing Task | F1 | ACC |
|---|---|---|---|
| QA | Unseen QA | **64.79** | **73.32** |
| | Translation | 51.34 | 65.10 |
| Translation | Unseen Translation | **74.03** | **73.81** |
| | QA | 20.45 | 37.50 |

Table 1: Zero-Shot Automatic Evaluation Results in the Same Task and across Different Tasks.

## 5 Results and Analysis

### 5.1 Results for Internal State-based Estimator

**(1) Seen/Unseen Query in Training Data** We evaluate our internal state-based estimator trained to distinguish unseen and seen questions. The F1 and accuracy scores reach 80.28% and 80.24%. These high results shed light on the effectiveness of our internal state-based method in identifying unseen queries. This phenomenon is aligned with the previous works (Kadavath et al., 2022; Yin et al., 2023b) which find the model can distinguish answerable and unanswerable questions that include future information.

**(2) Hallucination Risk faced with the Query** For estimating hallucination risk, as depicted in Fig. 3, our methods exhibit superior performance in both F1 and ACC. Notably, its performance remains stable across different tasks. It performs

| Task | Internal State | F1 | ACC |
|------|----------------|------|------|
| Dialogue | Llama2 | **74.33** | **74.22** |
| | Mistral | 72.39 | 72.55 |
| QA | Llama2 | **82.37** | **82.55** |
| | Mistral | 80.46 | 81.00 |
| Summarization | Llama2 | **88.08** | **88.95** |
| | Mistral | 83.63 | 85.42 |
| Translation | Llama2 | **76.90** | **76.90** |
| | Mistral | 73.10 | 73.14 |

Table 2: Automatic Evaluation Results of Internal States from Different Models.

less effectively in the translation task (F1 and ACC 76.90%) while excelling in the Number Conversion task (F1 94.04%, ACC 96.00%). Zero-shot prompt and ICL yield similar results, with ICL slightly outperforming zero-shot prompt. Both methods tend to be overconfident and predict LLM can accurately respond to the query (Recall 99%), which is aligned with the observation of (Xiong et al., 2023). PPL is better than the prompt methods while exhibiting varying performance across tasks. It performs poorly in the translation task (F1 33.73%, ACC 50.36%) but achieves its best performance in the data-to-text task (F1 88.28%, ACC 92.08%).

More results are described in Appendix B including treating separate metrics (Rouge-L, NLI, and QuestEval) as continuous regression labels and different estimator backbones.

## 5.2 Analysis

**Layer Depth *Positively* Correlates with its Prediction Performance.** We systematically dissect the contribution of each layer to the overall hallucination risk estimation. We hypothesize that certain layers may be more indicative of hallucinatory propensities than others, and our analysis seeks to validate this hypothesis. As shown in Fig. 4, early Layers perform poorly since they often capture basic syntactic information. Intermediate layers perform better since these layers typically encode more complex semantic relationships. Deep layers perform best and learn hallucination patterns with high-level presentation. This observation is different from Azaria and Mitchell (2023) where middle-layer hidden states of statements perform best in recognizing lying.

**Consistency of Internal States across Different LLMs** To evaluate the impact of the LLM's Internal State, we use Mistral-7B's internal state to

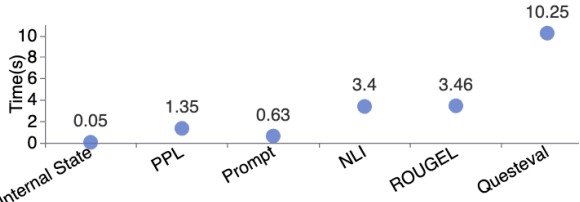

Figure 5: Inference time of various estimation methods.

assess Llama2's hallucination risk. As shown in Tab. 2, the results for four common NLG tasks exhibit a decrease compared to Llama2's own internal states. Since different LLMs share a similarity in model architecture and data, there is a potential for zero-shot transfer. Nonetheless, the most effective predictor of LLM's generative performance is still its own internal state, which underscores the importance of considering model-specific assessments rather than universal ones.

**Internal States Share Features inner-task but do not Cross-task.** As shown in Tab. 1, we evaluate the generalization across different NLG tasks and within the same NLG task. Specifically, we examine zero-shot performance in QA and translation. While the zero-shot performance within these individual tasks is acceptable, the cross-task generalization remains relatively weak, aligned with the findings reported by Kadavath et al. (2022).

In addition, we evaluate our estimator trained in QA on the out-of-domain hallucination QA dataset **ANAH** (Ji et al., 2024) to test our estimator's performance in the hallucination aspect and its generalization. ANAH is a bilingual dataset that offers analytical annotation of Hallucinations in LLMs within Generative QA. Our work uses English samples and treats the hallucination type as the label in the testing stage. The F1 score reaches 78.56% and the accuracy is 78.83%. These relatively high results shed light on the effectiveness of our internal state estimator in handling hallucination challenges and further show our generalization capabilities. Therefore, the features in internal states are shared with OOD data within the same task but not shared across tasks.

**Internal State as an *Efficient* Hallucination Estimator** Our estimator has three linear layers which requires minimal computing power. As shown in Fig. 5, our estimator demonstrates impressive efficiency. Specifically, likelihood-based costs 1.36s per sample, while internal-state-based costs only 0.05s per sample. This rapid inference

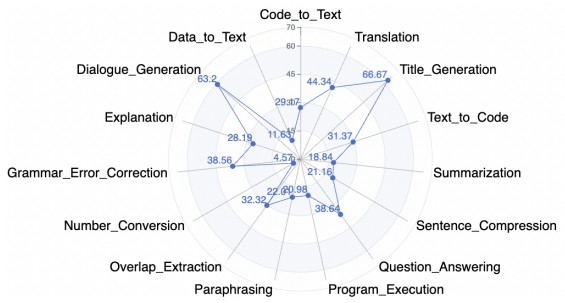

Figure 6: Hallucination Rate for each NLG Task.

speed is essential for real-world applications. The generation time, with a maximum token length of 50 and a batch size of 1, is 3.37 seconds. Notably, Questeval costs the most time 10.25s in total.

**Hallucination Rate** During the labeling process mentioned in § 3.2, we obtain the hallucination rate in the responses of each task. As illustrated in Fig. 6, the hallucination rate fluctuates significantly across NLG tasks. Among them, Title Generation exhibits the highest rate since its divergent nature and there is no unique and standard answer. In contrast, Number Conversion gains the lowest rate since the task is relatively easy and the answer is fixed leaving less room for hallucination.

**Visualizing Tokens Triggering Hallucination** To further understand the mechanisms behind hallucination, we dissect the process of the queries triggering hallucinations at a fine-grain level. Inspired by the Gradient-weighted Class Activation Mapping (Grad-CAM) technique (Selvaraju et al., 2017), we quantify the average gradients of input embedding associated with each token in the joint operation of the LLM and estimator. Specifically, we focused on how these tokens influence the LLM's internal state and the subsequent estimation of hallucinations based on this internal state.

Fig. 7 indicates that tokens within an unknown query contribute unequally to the occurrence of hallucinations. We observe that the tokens that are part of unfamiliar named entities or carry critical information exhibit a higher impact. For instance, "*amniotes*" in the QA task and "麻雀" in the translation task gain higher gradients and significantly impact hallucination estimation. This could be attributed to the system's attempts to generate fluent responses despite gaps in its understanding or knowledge about these entities.

**Error Analysis** Although our method performs better than the baselines in the estimation task, it

**Question:**
How many years ago did early amniotes diverge into two groups ?

**Golden Answer:**
320 million years

**Generated Reply:**
…occurred around 360-380 million years ago…

- - - - - - - - - - - - - - - - - - - - - - - - - - - - - -

**Translation Input:**
麻雀说服Tom用魔法指南针找到箱子。

**Golden Answer:**
The sparrow convinced Tom to use a magic compass to find the box.

**Generated Reply:**
Maggie convinced Tom to use a magic compass to find the box.

Figure 7: Visualization of Token Contributions to hallucinations in unknown queries for QA (top) and translation (bottom) tasks. Deeper background color means higher contributions and the hallucinated content in the generated reply is marked in pink .

still generates a few cases of failure. To gain more insight into our model, we present a failure example in Tab. 3 and conduct an error analysis. For example, the estimator predicts that LLM can correctly answer the query "*What is the term for tough, flexible connective tissue that contains the protein collagen?*" But LLM replies "*ligaments*" instead of "*cartilage*", which is hallucinated. the estimator predicts that LLM will hallucinate when faced with "*what appropriately nicknamed pacific location?*" But LLM replies without hallucination.

## 6 Related Work

**Knowledge Boundary** Researchers investigate the boundary of parametric knowledge in LLMs which aim to uncover *what models know and where their capabilities end*. Current research on the knowledge boundary predominantly narrows down this investigation to specialized tasks, charting a map of knowns and unknowns within these defined borders. In the representative QA field, Rajpurkar et al. (2018) and Yin et al. (2023b) collect datasets containing known and unknown questions and develop classifiers to differentiate them. The concept of unanswerability in these works is universal and *model-agnostic*, such as philosophical questions and unsolved mysteries. In contrast, Kadavath et al. (2022); Slobodkin et al. (2023), and Gottesman and Geva (2024) focus on *model-specific* prediction for questions. The former focuses on whether

| Query | Generated Reply | Golden Answer | Trainig Label | Predicted Label |
|---|---|---|---|---|
| Given a scientific question, generate a correct answer to it. What is the term for tough, flexible connective tissue that contains the protein collagen? | Ligaments are fibrous connective tissue structures that connect bones to other bones and provide support and stability to the joint... | cartilage | hallucinated | faithful |
| Given a scientific question, generate a correct answer to it. Since 1900, four of the five earthquakes of the greatest magnitude occurred near what appropriately nicknamed pacific location? | Since 1900, four of the five earthquakes of the greatest magnitude have occurred near the Pacific location of the Ring of Fire. The Ring of Fire is an area of... | ring of fire | faithful | hallucinated |

Table 3: Negative Samples. The hallucinated context is marked in pink.

the model will answer correctly, while the latter focuses on whether the model linguistically refuses to answer. Out-of-domain or out-of-distribution detection (Zhou et al., 2023; Ryu et al., 2018; Tan et al., 2019; Yang et al., 2021; Zheng et al., 2020) are also relevant areas dealing with the differentiation of unknown/unseen from training data, with main focus on classification tasks. Our method is versatile across various NLG tasks without requiring fine-tuning of LLMs.

**Hallucination Detection** The phenomenon of hallucination in NLG encourages a variety of detection methods (Min et al., 2023; Ji et al., 2024; Li et al., 2023; Scialom et al., 2021). Some of these methods delve into the internal states for detection. Azaria and Mitchell (2023), for example, collect a true-false statement dataset with *artificial guidance* and the classification results indicate that the LLMs' internal state can reveal the truthfulness of statements. INSIDE (Chen et al., 2024) also leverages LLMs' internal states and proposes EigenScore for evaluating the self-consistency of responses, thereby serving as a proxy for hallucination levels. MIND (Su et al., 2024), an unsupervised training approach, distinguishes the hallucinated continuation text from the original Wikipedia content based on internal states. Snyder et al. (2023) explore the query-only detection within the QA task based on internal states, gradients, and probabilities. On the other hand, Xiao and Wang (2021) shows evidence that higher uncertainty corresponds to a higher hallucination probability. Uncertainty estimation methods, such as (Xiao et al., 2022; Xiong et al., 2023; Kadavath et al., 2022), predict the reliability of their natural language outputs and can also serve as a tool for hallucination detection. Previous works leverage the LLM's internal states for the text to be measured which is not necessary from the same LLM. Differently, this work focuses on self-awareness corresponding to the queries across multiple NLG tasks.

## 7 Conclusion

Inspired by human self-awareness, this work demonstrates the latent capacity of LLMs to self-assess and estimate hallucination risks prior to response generation. We conduct a comprehensive analysis of the internal states of LLMs both in terms of training data sources and across 15 NLG tasks with over 700 datasets. Employing a probing estimator on the internal states associated with the queries, we assess their self-awareness and ability to indicate uncertainty in two aspects: (1) recognizing whether they have seen the query in training data, achieving an accuracy of 80.28%[5]. (2) recognizing whether they are likely to hallucinate when faced with the query. The results demonstrate that internal state-based self-assessment outperforms PPL-based and prompt-based baselines, with an average estimation accuracy of 84.32% across all tested datasets. In addition, we explore the role of particular neurons in uncertainty and hallucination perception and reveal a positive correlation between the depth of activation layers in an LLM and its predictive accuracy. The consistency of internal states across different models suggests a potential for zero-shot transfer, but model-specific estimation is the optimal strategy. Challenges of generalizing these findings across different tasks are noted, despite observing promising generalizations within the same NLG tasks.

For future work, we aim to refine our methodology to enhance the robustness and generalization across various NLG tasks in the field of hallucination risk assessment. In addition, we will involve a

---

[5]Please refer to the first parts of § 3.2 and § 5.1.

broader spectrum of LLMs to extend the applicability of our findings.

## 8 Limitation

**Model Coverage**   This work primarily investigate the widely used LLM, Llama2, due to its prevalence in current NLG applications. However, it does not encompass other LLMs. In the future, we will extend the scope of LLMs to enhance the robustness and applicability of our results.

**Human Evaluation in Data Construction**   Human judgment is extremely resource-intensive for hallucination judgment. The extensive time commitment and financial expenditure required are beyond the scope of this study, particularly given the large scale of the datasets. Consequently, this research did not include human evaluation in the data labeling process in § 3.2.

**Comparative Performance**   Our method predicts the risk in advance of generation and depends solely on the query. It may lead to a trade-off in performance compared to other existing approaches that consider both the query and the response.

## 9 Ethical Considerations

In our experiments, we utilized datasets that are either publicly accessible or synthetically generated, thereby circumventing any potential adverse effects on individuals or communities. The datasets employed in this investigation were meticulously curated and processed to uphold the principles of privacy and confidentiality. We ensured the exclusion of any personally identifiable information, with all data undergoing anonymization before any analysis was conducted.

When contemplating the deployment of our research outcomes, we recognize the inherent risks and ethical dilemmas involved. The tendency of LLMs to produce hallucinations could disproportionately affect various demographic groups, a consequence of the inherent biases in the training datasets. We are committed to the identification and rectification of such biases to forestall the continuation of stereotypes or the inequitable treatment of any demographic.

By adhering to these ethical considerations, we aim to contribute positively to the field of NLP and ensure that advancements in understanding and mitigating hallucinations in LLMs are achieved responsibly and with consideration for the broader societal impact.

## Acknowledgement

This work has been supported by the China NSFC Project NSFC21EG14 (No. 62120106006), SAAIR Project (No. Z1286), HKJCCT21EG01 (RG192), and Care-E Project (FS116).

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

## A  Dataset

This work uses benchmark Super-Natural Instructions (Wang et al., 2022b) which includes 1,616 diverse NLP datasets covering 76 distinct task types. We select 15 NLG task types and list all datasets included in each NLG task in Tab. A2.

## B  Results and Analysis

**Separate Metric as Continuous *Regression* Label**  In addition to the comprehensive integration of all metrics (i.e. NLI, Rouge-L, Questeval) described in § 3.2, we analyze our internal state-based method's performances when treating each metric as the label, separately.

We consider three forms of "golden score" for each metric. First, the absolute values, which serve as the target for regression, with higher scores indicating fewer hallucinations. We consider the probability of entailment as the absolute value of the NLI metric. Second, we standardize these absolute values using the minimum and maximum values from the training dataset to obtain normalized "golden scores". Third, we use the relative rankings of these scores within the training dataset as an alternative regression target.

For the regression task with continuous score predicted, we utilize **Root Mean Squared Error (RMSE)** to measure the average difference between the values predicted by our estimator and the actual values.

As shown in Fig. A1, our method's prediction performance varies across the form of the "golden score". For each metric, the RMSE is the smallest when predicting absolute value, which indicates that the hidden state performs best in predicting the absolute value of the metric. Conversely, the highest RMSE occurs when the model attempts to predict the relative rankings, implying that predicting the precise ordering of the metrics is more challenging for the hidden state representation.

**Estimator Backbone**  Instead of Llama MLP, we employ a standard MLP as the backbone of the estimator. The results in Tab. A1 demonstrate that Llama MLP outperforms the standard MLP.

## C  Implementation Details

The input dimension of our classifier is 4096 and the hidden dimension is 11008, which are aligned with Llama2-7B. We train our classifier with the following settings and hyper-parameters: the epoch

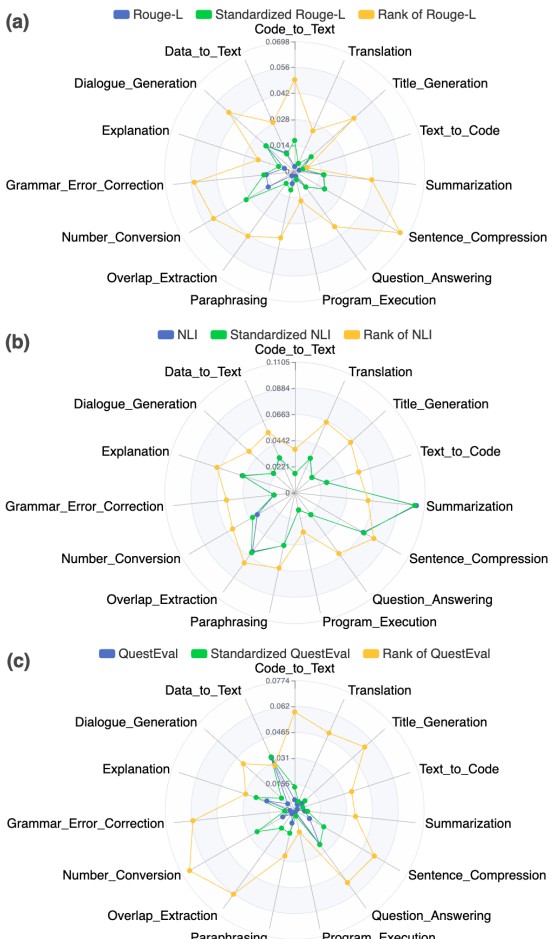

Figure A1: RMSE Scores of Internal State-based Estimator with Labels: (a) Rouge-L (b) NLI (c) QuestEval

| Task | Internal State | F1 | ACC |
|------|---------------|------|------|
| Dialogue | LlamaMLP | **74.33** | **74.22** |
| | MLP | 70.22 | 71.12 |
| QA | LlamaMLP | **82.37** | **82.55** |
| | MLP | 81.57 | 81.84 |
| Summarization | LlamaMLP | **88.08** | **88.95** |
| | MLP | 87.59 | 87.59 |
| Translation | LlamaMLP | **76.90** | **76.90** |
| | MLP | 74.23 | 74.90 |

Table A1: Automatic Evaluation Results for Different Classifier Backbone

is 10, the batch size is 128, the learning rate is 1e-5, and the AdamW optimizer has a linear scheduler. Our model is trained on 1 NVIDIA A800 GPU.

## D  AI Assistants Using

In this paper, we use ChatGPT to improve the writing at the grammar level.

| Task | No. | Dataset |
|------|-----|---------|
| Code to Text | 4 | Task 110: logic2text sentence generation, Task 129: scan long text generation action command short, Task 127: scan long text generation action command all, Task 131: scan long text generation action command long |
| Data to Text | 9 | Task 1728: web nlg data to text, Task 1598: nyc long text generation, Task 1631: openpi answer generation, Task 677: ollie sentence answer generation, Task 957: e2e nlg text generation generate, Task 760: msr sqa long text generation, Task 1407: dart question generation, Task 102: commongen sentence generation, Task 1409: dart text generation |
| Dialogue Generation | 13 | Task 574: air dialogue sentence generation, Task 361: spolin yesand prompt response classification, Task 576: curiosity dialogs answer generation, Task 1603: smcalflow sentence generation, Task 1714: convai3 sentence generation, Task 1730: personachat choose next, Task 565: circa answer generation, Task 611: mutual multi turn dialogue, Task 1729: personachat generate next, Task 1600: smcalflow sentence generation, Task 639: multi woz user utterance generation, Task 1590: diplomacy text generation, Task 360: spolin yesand response generation |
| Explanation | 6 | Task 295: semeval 2020 Task 4: commonsense reasoning, Task 192: hotpotqa sentence generation, Task 593: sciq explanation generation, Task 1369: healthfact sentence generation, Task 223: quartz explanation generation, Task 134: winowhy reason generation |
| Grammar Error Correction | 2 | Task 1415: youtube caption corrections grammar correction, Task 1557: jfleg answer generation |
| Number Conversion | 2 | Task 1703: ljspeech textmodification, Task 1704: ljspeech textmodification |
| Overlap Extraction | 2 | Task 039: qasc find overlapping words, Task 281: points of correspondence |
| Paraphrasing | 12 | Task 776: pawsx japanese text modification, Task 045: miscellaneous sentence paraphrasing, Task 770: pawsx english text modification, Task 771: pawsx korean text modification, Task 774: pawsx german text modification, Task 177: para-nmt paraphrasing, Task 466: parsinlu qqp text modification, Task 775: pawsx chinese text modification, Task 1614: sick text modify, Task 773: pawsx spanish text modification, Task 132: dais text modification, Task 772: pawsx french text modification |
| Program Execution | 90 | Task 113: count frequency of letter, Task 1151: swap max min, Task 509: collate of all alphabetical and numerical elements in list separately, Task 100: concatenate all elements from index i to j, Task 096: conala list index subtraction, Task 365: synthetic remove vowels, Task 622: replace alphabets in a list by their position in english alphabet, Task 852: synthetic multiply odds, Task 1088: array of products, Task 1405: find median, Task 637: extract and sort unique digits in a list, Task 1446: farthest integers, Task 506: position of all alphabetical elements in list, Task 378: reverse words of given length, Task 093: conala normalize lists, Task 1404: date conversion, Task 097: conala remove duplicates, Task 372: synthetic palindrome numbers, Task 755: find longest substring and replace its sorted lowercase version in both lists, Task 636: extract and sort unique alphabets in a list, Task 267: concatenate and reverse all elements from index i to j, Task 162: count words starting with letter, Task 159: check frequency of words in sentence pair, Task 208: combinations of list, Task 1316: remove duplicates string, Task 504: count all alphabetical elements in list, Task 079: conala concat strings, Task 158: count frequency of words, Task 507: position of all numerical elements in list, Task 374: synthetic pos or neg calculation, Task 1087: two number sum, Task 163: count words ending with letter, Task 756: find longert substring and return all unique alphabets in it, Task 101: reverse and concatenate all elements from index i to j, Task 1551: every ith element from kth element, Task 606: sum of all numbers in list between positions i and j, Task 368: synthetic even or odd calculation, Task 1150: delete max min, Task 851: synthetic multiply evens, Task 377: remove words of given length, Task 063: first i elements, Task 064: all elements except first i, Task 245: check presence in set intersection, Task 161: count words containing letter, Task 605: find the longest common subsequence in two lists, Task 850: synthetic longest palindrome, Task 157: count vowels and consonants, Task 373: synthetic round tens place, Task 206: collatz conjecture, Task 1443: string to number, Task 123: conala sort dictionary, Task 244: count elements in set union, Task 499: extract and add all numbers from list, Task 124: conala pair averages, Task 1444: round power of two, Task 099: reverse elements between index i and j, Task 1089: check monotonic array, Task 1188: count max freq char, Task 125: conala pair differences, Task 488: extract all alphabetical elements from list in order, Task 1542: every ith element from starting, Task 1194: kth largest element, Task 371: synthetic product of list, Task 1406: kth smallest element, Task 095: conala max absolute value, Task 1315: find range array, Task 243: count elements in set intersection, Task 1331: reverse array, Task 062: bigbench repeat copy logic, Task 122: conala list index addition, Task 091: all elements from index i to j, Task 369: synthetic remove odds, Task 497: extract all numbers from list in order, Task 505: count all numerical elements in list, Task 205: remove even elements, Task 1189: check char in string, Task 1445: closest integers, Task 094: conala calculate mean, Task 160: replace letter in a sentence, Task 1148: maximum ascii value, Task 098: conala list intersection, Task 078: all elements except last i, Task 523: find if numbers or alphabets are more in list, Task 370: synthetic remove divisible by 3, Task 367: synthetic remove floats, Task 1190: add integer to list, Task 376: reverse order of words, Task 600: find the longest common substring in two strings, Task 207: max element lists, Task 366: synthetic return primes |

| Task | No. | Dataset |
|------|-----|---------|
| Question Answering | 207 | Task 837: viquiquad answer generation, Task 701: mmmlu answer generation high school computer science, Task 1399: obqa answer generation, Task 075: squad1.1 answer generation, Task 724: mmmlu answer generation moral scenarios, Task 666: mmmlu answer generation astronomy, Task 742: lhoestq answer generation frequency, Task 1438: doqa cooking answer generation, Task 863: asdiv multiop question answering, Task 864: asdiv singleop question answering, Task 058: multirc question answering, Task 669: ambigqa answer generation, Task 704: mmmlu answer generation high school government and politics, Task 728: mmmlu answer generation professional accounting, Task 740: lhoestq answer generation quantity, Task 1293: kilt tasks hotpotqa question answering, Task 849: pubmedqa answer generation, Task 1424: mathqa probability, Task 1625: disfl qa asnwer generation, Task 858: inquisitive span detection, Task 723: mmmlu answer generation moral disputes, Task 083: babi t1 single supporting fact answer generation, Task 118: semeval 2019 Task 10: open vocabulary mathematical answer generation, Task 582: naturalquestion answer generation, Task 237: iirc answer from subtext answer generation, Task 714: mmmlu answer generation human sexuality, Task 444: com qa question paraphrases answer generation, Task 720: mmmlu answer generation marketing, Task 332: tellmewhy answer generation, Task 119: semeval 2019 Task 10: geometric mathematical answer generation, Task 310: race classification, Task 1132: xcsr ur commonsense mc classification, Task 702: mmmlu answer generation high school european history, Task 710: mmmlu answer generation high school statistics, Task 870: msmarco answer generation, Task 047: miscellaneous answering science questions, Task 711: mmmlu answer generation high school us history, Task 1286: openbookqa question answering, Task 598: cuad answer generation, Task 685: mmmlu answer generation clinical knowledge, Task 084: babi t1 single supporting fact identify relevant fact, Task 1420: mathqa general, Task 1520: qa srl answer generation, Task 868: mawps singleop question answering, Task 768: qed text span selection, Task 061: ropes answer generation, Task 041: qasc answer generation, Task 144: subjqa question answering, Task 1570: cmrc2018 answer generation, Task 1610: xquad es answer generation, Task 164: mcscript question answering text, Task 703: mmmlu answer generation high school geography, Task 705: mmmlu answer generation high school macroeconomics, Task 1131: xcsr es commonsense mc classification, Task 1130: xcsr vi commonsense mc classification, Task 750: aqua multiple choice answering, Task 473: parsinlu mc classification, Task 385: socialiqa incorrect answer generation, Task 691: mmmlu answer generation college physics, Task 719: mmmlu answer generation management, Task 1327: qa zre answer generation from question, Task 715: mmmlu answer generation international law, Task 737: mmmlu answer generation world religions, Task 010: mctaco answer generation event ordering, Task 741: lhoestq answer generation place, Task 028: drop answer generation, Task 730: mmmlu answer generation professional medicine, Task 491: mwsc answer generation, Task 716: mmmlu answer generation jurisprudence, Task 732: mmmlu answer generation public relations, Task 735: mmmlu answer generation us foreign policy, Task 898: freebase qa answer generation, Task 887: quail answer generation, Task 024: cosmosqa answer generation, Task 1140: xcsr pl commonsense mc classification, Task 225: english language answer generation, Task 1608: xquad en answer generation, Task 170: hotpotqa answer generation, Task 667: mmmlu answer generation business ethics, Task 699: mmmlu answer generation high school biology, Task 595: mocha answer generation, Task 751: svamp subtraction question answering, Task 1656: gooaq answer generation, Task 1431: head qa answer generation, Task 1296: wiki hop question answering, Task 490: mwsc options generation, Task 867: mawps multiop question answering, Task 865: mawps addsub question answering, Task 1133: xcsr nl commonsense mc classification, Task 1422: mathqa physics, Task 1135: xcsr en commonsense mc classification, Task 054: multirc write correct answer, Task 1661: super glue classification, Task 708: mmmlu answer generation high school physics, Task 1726: mathqa correct answer generation, Task 664: mmmlu answer generation abstract algebra, Task 1412: web questions question answering, Task 002: quoref answer generation, Task 752: svamp multiplication question answering, Task 1297: qasc question answering, Task 692: mmmlu answer generation computer security, Task 1136: xcsr fr commonsense mc classification, Task 727: mmmlu answer generation prehistory, Task 725: mmmlu answer generation nutrition, Task 104: semeval 2019 Task 10: closed vocabulary mathematical answer generation, Task 694: mmmlu answer generation econometrics, Task 820: protoqa answer generation, Task 700: mmmlu answer generation high school chemistry, Task 390: torque text span selection, Task 1421: mathqa other, Task 918: coqa answer generation, Task 309: race answer generation, Task 247: dream answer generation, Task 695: mmmlu answer generation electrical engineering, Task 230: iirc passage classification, Task 712: mmmlu answer generation high school world history, Task 731: mmmlu answer generation professional psychology, Task 596: mocha question generation, Task 698: mmmlu answer generation global facts, Task 718: mmmlu answer generation machine learning, Task 395: persianqa answer generation, Task 597: cuad answer generation, Task 339: record answer generation, Task 835: mathdataset answer generation, Task 238: iirc answer from passage answer generation, Task 228: arc answer generation easy, Task 380: boolq yes no question, Task 152: tomqa find location easy noise, Task 754: svamp common-division question answering, Task 713: mmmlu answer generation human aging, Task 665: mmmlu answer generation anatomy, Task 706: mmmlu answer generation high school mathematics, Task 697: mmmlu answer generation formal logic, Task 753: svamp addition question answering, Task 1727: wiqa what is the effect, Task 1139: xcsr ru commonsense mc classification, Task 1134: xcsr hi commonsense mc classification, Task 344: hybridqa answer generation, Task 165: mcscript question answering commonsense, Task 1145: xcsr jap commonsense mc classification, Task 1295: adversarial qa question answering, Task 239: tweetqa answer generation, Task 1382: quarel write correct answer... |

| Task | No. | Dataset |
|------|-----|---------|
| Translation | 394 | Task 808: pawsx chinese korean translation, Task 254: spl translation fi en, Task 1111: ted translation he it, Task 988: pib translation oriya english, Task 650: opus100 ar en translation, Task 763: emea es lt translation, Task 1648: opus books en-sv translation, Task 1263: ted translation pl fa, Task 1020: pib translation telugu oriya, Task 913: bianet translation, Task 1060: pib translation urdu malayalam, Task 1676: xquad-ca translation, Task 1098: ted translation ja fa, Task 984: pib translation marathi gujarati, Task 1086: pib translation marathi english, Task 789: pawsx french english translation, Task 1110: ted translation he gl, Task 1689: qed amara translation, Task 787: pawsx korean chinese translation, Task 1071: pib translation malayalam marathi, Task 548: alt translation en ch, Task 1373: newscomm translation, Task 1023: pib translation english hindi, Task 1271: ted translation fa it, Task 1274: ted translation pt en, Task 552: alt translation en bu, Task 1040: pib translation punjabi oriya, Task 1323: open subtitles hi en translation, Task 1058: pib translation urdu english, Task 1105: ted translation ar gl, Task 1353: hind encorp translation en hi, Task 1085: pib translation english marathi, Task 1103: ted translation es fa, Task 784: pawsx korean french translation, Task 811: pawsx chinese german translation, Task 1365: opustedtalks translation, Task 1278: ted translation pt he, Task 1115: alt ja id translation, Task 538: alt translation bu en, Task 786: pawsx korean german translation, Task 805: pawsx german chinese translation, Task 1692: qed amara translation, Task 1690: qed amara translation, Task 655: bible en fa translation, Task 1256: ted translation pl en, Task 977: pib translation oriya urdu, Task 841: para pdt de en translation, Task 996: pib translation english bengali, Task 531: europarl es en translation, Task 452: opus paracrawl en ig translation, Task 1250: ted translation it ar, Task 644: refresd translation, Task 1248: ted translation it ja, Task 1034: pib translation hindi gujarati, Task 1225: ted translation ja he, Task 997: pib translation bengali oriya, Task 1127: alt ja th translation, Task 783: pawsx korean english translation, Task 1031: pib translation bengali telugu, Task 560: alt translation en entk, Task 1000: pib translation tamil malayalam, Task 252: spl translation en tr, Task 1650: opus books en-fi translation, Task 654: bible fa en translation, Task 802: pawsx german korean translation, Task 1025: pib translation bengali punjabi, Task 262: spl translation ja en, Task 785: pawsx korean spanish translation, Task 530: europarl en es translation, Task 1232: ted translation ar es, Task 799: pawsx spanish chinese translation, Task 1119: alt fil ja translation, Task 260: spl translation zh en, Task 1686: menyo20k translation, Task 448: opus paracrawl en tl translation, Task 994: pib translation tamil hindi, Task 1065: pib translation punjabi telugu, Task 557: alt translation en ba, Task 1072: pib translation marathi malayalam, Task 535: alt translation ch en, Task 762: emea fr sk translation, Task 1024: pib translation hindi english, Task 914: bianet translation, Task 779: pawsx english spanish translation, Task 547: alt translation entk en, Task 1128: alt th ja translation, Task 537: alt translation th en, Task 1277: ted translation pt ar, Task 1124: alt ja lo translation, Task 1514: flores translation entone, Task 435: alt en ja translation, Task 425: hindienglish corpora en hi translation, Task 1371: newscomm translation, Task 818: pawsx japanese chinese translation, Task 873: opus xhosanavy translation xhosa eng, Task 1240: ted translation gl es, Task 553: alt translation en ma, Task 1351: opus100 translation gu en, Task 999: pib translation malayalam tamil, Task 438: eng guj parallel corpus en gu translation, Task 541: alt translation kh en, Task 1329: open subtitles en hi translation, Task 1102: ted translation es pl, Task 661: mizan en fa translation, Task 1259: ted translation pl ar, Task 424: hindienglish corpora hi en translation, Task 793: pawsx french chinese translation, Task 1005: pib translation malayalam punjabi, Task 1262: ted translation pl it, Task 1367: opustedtalks translation, Task 117: spl translation en de, Task 1237: ted translation he ar, Task 1122: alt khm ja translation, Task 1230: ted translation ar en, Task 790: pawsx french korean translation, Task 433: alt hi en translation, Task 253: spl translation en zh, Task 1037: pib translation telugu urdu, Task 840: para pdt en es translation, Task 982: pib translation tamil bengali, Task 1009: pib translation bengali hindi, Task 1062: pib translation marathi bengali, Task 1218: ted translation en ja, Task 1113: ted translation he fa, Task 1691: qed amara translation, Task 1276: ted translation pt es, Task 1108: ted translation ar fa, Task 1070: pib translation urdu bengali, Task 1244: ted translation gl pl, Task 1239: ted translation gl ja, Task 1055: pib translation marathi oriya, Task 794: pawsx french japanese translation, Task 1004: pib translation malayalam bengali, Task 1049: pib translation malayalam telugu, Task 989: pib translation marathi urdu, Task 450: opus paracrawl so en translation, Task 815: pawsx japanese french translation, Task 1066: pib translation telugu punjabi, Task 777: pawsx english korean translation, Task 542: alt translation ja en, Task 830: poleval2019 mt translation, Task 1655: mkb translation, Task 313: europarl en sv translation, Task 1044: pib translation punjabi gujarati, Task 1038: pib translation urdu telugu, Task 1057: pib translation english urdu, Task 1047: pib translation english telugu, Task 1258: ted translation pl es, Task 1001: pib translation gujarati urdu, Task 1063: pib translation gujarati tamil, Task 1649: opus books en-no translation, Task 1282: ted translation pt fa, Task 983: pib translation gujarati marathi, Task 261: spl translation es en, Task 439: eng guj parallel corpus gu en translation, Task 795: pawsx spanish english translation, Task 1046: pib translation telugu hindi, Task 1233: ted translation ar he, Task 1112: ted translation he pl, Task 663: global voices en fa translation, Task 662: global voices fa en translation, Task 1376: newscomm translation, Task 258: spl translation fa en, Task 1029: pib translation marathi punjabi, Task 986: pib translation oriya hindi, Task 1067: pib translation bengali gujarati, Task 604: flores translation entosn, Task 1224: ted translation ja ar, Task 250: spl translation en ar, Task 1242: ted translation gl he, Task 559: alt translation en fi, Task 1015: pib translation punjabi tamil, Task 259: spl translation tr en, Task 1269: ted translation fa he, Task 807: pawsx chinese english translation, Task 809: pawsx chinese french translation, Task 995: pib translation bengali english, Task 1093: ted translation en fa, Task 174: spl translation en ja, Task 1036: pib translation urdu tamil... |

| Task | No. | Dataset |
|------|-----|---------|
| Sentence Compression | 1 | Task 1340: msr text compression compression |
| Summarization | 16 | Task 1357: xlsum summary generation, Task 672: amazon and yelp summarization dataset summarization, Task 1579: gigaword incorrect summarization, Task 1658: billsum summarization, Task 618: amazonreview summary text generation, Task 522: news editorial summary, Task 1355: sent comp summarization, Task 589: amazonfood summary text generation, Task 1553: cnn dailymail summarization, Task 1572: samsum summary, Task 1291: multi news summarization, Task 668: extreme abstract summarization, Task 1309: amazonreview summary classification, Task 1499: dstc3 summarization, Task 1290: xsum summarization, Task 511: reddit tifu long text summarization |
| Text to Code | 12 | Task 210: logic2text structured text generation, Task 107: splash question to sql, Task 077: splash explanation to sql, Task 076: splash correcting sql mistake, Task 130: scan structured text generation command action long, Task 869: cfq mcd1 sql to explanation, Task 212: logic2text classification, Task 126: scan structured text generation command action all, Task 211: logic2text classification, Task 128: scan structured text generation command action short, Task 868: cfq mcd1 explanation to sql, Task 956: leetcode 420 strong password check |
| Title Generation | 19 | Task 1540: parsed pdfs summarization, Task 1561: clickbait new bg summarization, Task 769: qed summarization, Task 1342: amazon us reviews title, Task 1356: xlsum title generation, Task 569: recipe nlg text generation, Task 1161: coda19 title generation, Task 220: rocstories title classification, Task 219: rocstories title answer generation, Task 1586: scifact title generation, Task 602: wikitext-103 answer generation, Task 1358: xlsum title generation, Task 1659: title generation, Task 418: persent title generation, Task 743: eurlex summarization, Task 288: gigaword summarization, Task 500: scruples anecdotes title generation, Task 619: ohsumed abstract title generation, Task 510: reddit tifu title summarization |

Table A2: Dataset list for each NLG task from Super-Natural Instructions.