# OpenReview forum: "LLM Internal States Reveal Hallucination Risk Faced With a Query"
_EMNLP/2024/Workshop/BlackBoxNLP — BlackboxNLP 2024_

### Official Review · Reviewer_pzw4 · 2024-09-08

**Overall Assessment:** 3
**Confidence:** 3

**Best Paper:**

1

**Best Paper Justification:**

none

**Comments Questions Suggestions And Typos:**

Do not use contractions like "don't" and "it's".
Clarify the visualizations in Figure 2 and what they show.
Better explain the selection of internal states that you use in your classifier.

**Paper Summary:**

The paper investigates the use of internal LLM states for the prediction of hallucination risks. The motivation is that generative models can assess how much they can reliably generate answers to given queries and in this way, hallucinations can be reduced. The approach uses a MLP classifier to estimate hallucination risk and the method is tested on a large variety of tasks.

**Summary Of Strengths:**

Making it possible to assess internal states in a simple and straightforward manner to detect the risk of hallucinations is very useful in real-world applications. The approach seems to be robust and reliable compared to other baseline approaches. The method has been tested on a wide variety of tasks and the results look convincing.

**Summary Of Weaknesses:**

Some decisions look a bit ad-hoc to me and would require more detailed explanations. In particular, the preliminary study in section 3.3 seems to be crucial for the selection of the internal states that are used as input for the estimator trained below. I do not clearly see how this analysis supports the decision of taking the internal states corresponding to the last tokens in a query as stated in 3.4. Where exactly comes the evidence for this particular selection? I am also not really sure how to read figure 2 and how that would help me to understand the selection of states to be taken for your classifier.

In the data construction, it is good to see that you combine different automatic metrics to label the data. Nevertheless, I would like to see a discussion on the reliability of this kind of annotation and the particular combination heuristics that you apply.

In some tasks, I also wonder how you distinguish between seen and unseen examples. For example in translation, what does it mean to query for unseen translations? The same question arises for other tasks.

Personally, I would also tone down the analogies with human cognition and self-awareness. It is nice to see that self-assessment can be implemented to improve uncertainty estimation in prediction but the comparison to human self-awareness and cognitive processes stretches the ability a bit too far. Instead, I would focus on the technical aspects of model calibration and uncertainty estimation that are practical features for more reliable and trustworthy output generation. No need to draw a picture of conscious and self-aware LLMs.

---

### Official Review · Reviewer_zuHT · 2024-09-11

**Overall Assessment:** 3
**Confidence:** 3

**Best Paper:**

1

**Comments Questions Suggestions And Typos:**

Missing references:

Slobodkin, A., Goldman, O., Caciularu, A., Dagan, I., & Ravfogel, S. (2023, December). The curious case of hallucinatory (un) answerability: Finding truths in the hidden states of over-confident large language models. In Proceedings of the 2023 Conference on Empirical Methods in Natural Language Processing (pp. 3607-3625).

Gottesman, D., & Geva, M. (2024). Estimating Knowledge in Large Language Models Without Generating a Single Token. arXiv preprint arXiv:2406.12673.

Dataset section under Experiment section is repitative. Consider merging it with the "Data Construction" setction

Note that sometimes a period is missing at the end of a sentence after using numbers. for example line 364: " [...] accuracy scores reach 80.28% and 80.24% These"

**Paper Summary:**

This paper entitled "LLM Internal States Reveal Hallucination Risk Faced With a Query" explore the ability of LLMs' internals to reveal a tendency to hallucinate, even at the query stage.

The authors examine their question by two main types of datasets -
(1) Questions regarding news from the past year (which the model was not exposed to, since it is after the training point) versus old news
(2) Collection of 15 NLG dataset automatically annotated for hallucinations risks

The authors compared the internal state of correct answers vs wrong.

The authors conclude that: LLM internal states indicate (1) if query was on training data, and (2) if they likely to hallucinate

**Summary Of Strengths:**

The topic is important.

The authors performed interesting experiments that can contribute to understanding of LLMs mechanisms.

The authors proposed diverse visualizations.

I liked the experiment that related to efficiency of methods that highlight the inference time of the different methods.

**Summary Of Weaknesses:**

My main criticism is for the level of documentation, over claiming regarding conclusions and the experiment setup was not satisfactory - it is not entirely clear what was carried out, lacking details. I'm not sure that the authors measuring "hallucinations" in their experiments (see my comments below).

In addition, there was a lack of reference to relevant literature  (see examples in "missing references" under comments and suggestions).
How does this work add to previous existing literature?

Questions and comments in chronological order:
Figure 2 - What is the meaning of the space (the positions of the points on the x-axis and y-axis) ?

"Seen/Unseen Query in Training" dataset:
I did not understand how the authors determined whether the models had seen the query in training data. Is there a report reporting this?
For the "seen group", why is  LLM likely exposed during training to historical BBC news in 2020?
On the other hand, how could you be sure that the models was not exposed to the latest news?

"Hallucination Risk" dataset:
I am not sure it measures what the authors want it to measure. The dataset represents the automatic measures and not the hallucinations. The authors should consider at least reporting for a subset of the dataset about manual-human evaluation for hallucinations.

Problem formulation (Section 3.1) wasn't clear to me. E.g., h have two different meanings

The results for the " Seen/Unseen" are not clear. What the numbers exactly represents? Why should F1 and accuracy be reported?
What have been train? how? A lot of details are missing.
I'm not convinced that the model the authors built knows how to identify questions it knows or doesn't know the answer to, but maybe it simply recognizes a text from before 2021 versus a text that happened after 2024.

With respect to "Hallucination Risk", in Figure 3 consider adding the automatic measures that the LLMs' used to be train on.

What does the experiment shown in Figure 4 teach us? It seems quite trivial that as the model progresses through the layers it will converge more and more to the correct answer.

The authors offer an error analysis but it is actually anecdotal examples without conclusions

---

### Official Review · Reviewer_QrEr · 2024-09-12

**Overall Assessment:** 3
**Confidence:** 3

**Best Paper:**

1

**Best Paper Justification:**

N/A

**Comments Questions Suggestions And Typos:**

- The use of "self-awareness", and references to inspiration from human self-awareness relate to the status of LLM consciousness, which might be considered to be out of the scope of this work, and best adjudicated elsewhere
- When the three labelling methods (ROUGE-L, NLI, QuestEval) are ensembled, the rule seems quite conservative in treating the label as 0 if any of the methods labels it 0. Is there evidence that this strategy is appropriate?

**Paper Summary:**

The paper presents a method to probe a language model's internal state representation to detect hallucinations. The paper considers two settings: hallucinations as generations not supported in the training data, and hallucinations as detected in the ANAH framework. The paper finds that the probe is able to correctly label hallucinated generations with high accuracy/F1 score, higher than prompting and perplexity-based baselines. The paper also presents analyses of the influence of layer depth, probe generalization, and the efficiency of the proposed method for detecting hallucinations.

**Summary Of Strengths:**

- The proposed approach is well-motivated
- The proposed approach is shown to work in a variety of NLG settings
- The proposed approach is shown to beat strong non-probing baselines

**Summary Of Weaknesses:**

- The way the hallucination risk labels are obtained is unclear, and it is not apparent what the probe is detecting in experiment 2
- The validity of the labels for the probes is not independently assessed – how are we to know that these probe labels, that are automatically generated, are appropriate?
- It is unclear how figure 2 is interpreted to be showing the separability of hallucinated generations
- Probing method typically need to be contextualized with results from control tasks [1], which don't seem to be discussed, and would help understand the presented accuracy numbers better


[1] John Hewitt and Percy Liang. 2019. Designing and Interpreting Probes with Control Tasks. In Proceedings of the 2019 Conference on Empirical Methods in Natural Language Processing and the 9th International Joint Conference on Natural Language Processing (EMNLP-IJCNLP), pages 2733–2743, Hong Kong, China. Association for Computational Linguistics.

---

### Decision · Program_Chairs · 2024-09-19

**Decision:**

Accept

**Comment:**

Reviewers agree on the importance of the topic and find the approach robust. However, there are some concerns about the reliability/validity of the labels/annotations and the overclaims in the paper. I accept the paper with the borderline score and highly encourage the authors to take the reviewers' comments into account to improve the writing and presentation of the paper, try to clarify the visualizations and detail experimental designs, incorporate suggested references, moderate the tone when drawing conclusions, and be more careful with the terminology used in the paper for the camera-ready version.